# Eye Infection with SARS-CoV-2 as a Route to Systemic Immunization?

**DOI:** 10.3390/v14071447

**Published:** 2022-06-30

**Authors:** Norbert Schrage, Joel Blomet, Frank Holzer, A. Tromme, F. Ectors, Daniel Desmecht

**Affiliations:** 1ACTO e.V., Karlsburgweg 9, 52070 Aachen, Germany; 2Laboratoires Prevor, Moulin de Verville, 95670 Valmondois, France; jblomet@prevor.com; 3Ursapharm Arzneimittel GmbH, Industriestraße 35, 66129 Saarbrücken, Germany; dorothea.gross@ursapharm.de; 4CHU Liege, Clinique Veterinaire Universitaire, Quartier Vallee 2 Avenue de Cureghem 3, Sart-Tilman, 4000 Liege, Belgium; atromme@uliege.be (A.T.); Fabien.ectors@uliege.be (F.E.); daniel.desmecht@uliege.be (D.D.)

**Keywords:** SARS-CoV-2, mucosa, MALT, conjunctiva, SARS-CoV-2 conjunctivitis, immunization

## Abstract

Infectious diseases of the conjunctiva and cornea usually leave behind both broad local and systemic immunity. Case reports of SARS-CoV-2-positive conjunctivitis with subsequent systemic immunity suggest a new route of immunization preventing the primary infection of the airways. Material and Methods: A total of 24 Syrian field hamsters were treated. In systematic animal experiments, we infected the eyes of *n* = 8 animals (group 1) and the airways of another *n* = 8 animals (group 2) with SARS-CoV-2 (Wuhan type); *n* = 8 hamsters served as controls (group 3). The weight development of the animals was recorded. After two weeks of observation of disease symptoms, all animals were re-exposed to SARS-CoV-2 in the respiratory tract (challenge) to determine whether immunity to the virus had been achieved. Results: The epi-ocularly infected animals (group 1) showed no clinically visible disease during the ocular infection phase. At most, there was a slightly reduced weight gain compared to the control group (group 3), while the respiratory infected animals (group 2) all lost weight, became lethargic, and slowly recovered after two weeks. After the challenge, none of the animals in groups 1 and 2 became ill again. The animals in the negative control (group 3) all became ill. Cytotoxic antibodies were detectable in the blood of the infected groups before and after challenge, with higher titers in the epi-ocularly infected animals. Conclusion: By epi-ocular infection with SARS-CoV-2, the development of systemic immunity with formation of cytotoxic antibodies without severe general disease could be observed in the experimental animals, which did not induce any more disease upon a second infection in the respiratory tract. Therefore, it can be concluded that a purely epi-ocular infection with SARS-CoV2 only induces a weak disease pattern followed by systemic immunity.

## 1. Introduction

Severe acute respiratory syndrome corona virus 2 (SARS-CoV-2) has been a threat to general health systems after more than half a billion cases and more than 6 million cumulative deaths [1]. The development and spread of the disease, and its biological and socioeconomic background, are of major impact for new scientific driven innovations and modifications of health systems [2]. Since 2020, intramuscular vaccinations have been spread throughout the world and induce strong long lasting serum immunity [3]. Nevertheless, previous studies suggested that systemic immunity to severe systemic disease can be induced by infection of an organ not described as the primary airway targets such as the nose, throat, and lung [4,5]. There have been publications recording human infections with isolated SARS-CoV-2 positive conjunctivitis in Saudi and Turkish colleagues without any general disease during the 2019–2022 SARS-CoV-2. This type of infection suggests a special conclusion that an ocular infection will result in a systemic immunization [6,7]. If isolated infection of the eye with SARS-CoV-2 leads to a systemic immune response, this finding could be useful in innovative approaches to combat the pandemic, especially due to the fact that an immunizing SARS-CoV2 infection can take place at another site of the boy, excluding the normal transmission mechanism of breathing and thus preventing spreading and autoinfection of the lungs. Furthermore, the eye is part of the mucosa adherent lymphatic tissue (MALT System) which transfers and expresses mucosal and humoral immunity all over the body [4]. The receptor status of the eye has currently been identified as suitable for the SARS-CoV-2 virus [8]. Could this also be used to deliver vaccines via the conjunctival tissue to evoke systemic immunization? Our hypothesis to prove in the presented scientific experiment is: that a primary conjunctival infection with the virus will allow the body to develop immunity before the arrival of virus in the nose via the naso-lacrymal duct. Why do we believe this? Ophthalmologists know epidemic keratoconjunctivitis by adeno-pharyngeal conjunctival virus (APC) as a highly infectious viral droplet and smear infection. The infection leads to general immunization against the specific virus. Typically, epidemic keratoconjunctivitis is caused by adenoviruses and is thus an example of systemic immunization via the eyes [5]. With regard to SARS-CoV-2, the presumed mechanism of isolated infection of the eye as an example of low-risk immunization can initially only be tested experimentally in live animals. That there is an effective systemic immunization via the conjunctiva is shown by the experiments on epi-ocular vaccination with Salmonella typhimurum and influenza viruses in mice [9]. Impressive results on this were already published in 2015 by Eun-Do Kim et al. [10]. A Chinese research group investigated whether immunization against SARS-CoV-2 via the eye is now possible in rhesus monkeys. After eye infection with SARS-CoV-2, the monkeys developed mild pneumonia but no severe general disease [11]. The next step is now to prove immunization by isolated infection of the eyes in animal experiments.

## 2. Materials and Methods

A total of 24 female Syrian hamsters were selected as experimental animals [12]. The hamsters, approximately 5 weeks old and homogeneous in weight, were purchased from the Janvier company and underwent a clinical examination by two experienced veterinarians upon arrival at the laboratory. The delivered animals showed no atypical posture, no particular behavior, and no skin lesions. The animals were housed in an SPF animal facility for one week to mitigate the possible effects of transport stress. Subsequently, the animals were relocated to an A3 facility.

### 2.1. Experimental Groups

Animals were divided and treated into the following groups for the experiment: 

Group 1 (G1) (n = 5), intranasally exposed as a positive control group. Animals were intranasally infected with 3 × 10^6^ SARS-CoV-2 viruses under Isofluran^®^ (Baxter Germany) GmbH, Edisonstraße 4, 85716 Unterschleißheim anesthesia (5%/95% oxygen) on day 0 and euthanized after 21 days.

Group 2 (G2) (n = 7), exposed intranasally with challenge. Animals were intranasally infected with 3 × 10^6^ SARS-CoV-2 viruses under Isofluran^®^ anesthesia (5%/95% oxygen) on day 0. After 21 days, the animals were exposed to a new intranasal infection with 3 × 10^6^ SARS-CoV-2 viruses at the same dose (challenge) and euthanized 7 days after challenge at day 28. 

The inoculum SARS-CoV-2 virus (strain UCH Liège [13]) was instilled into the nostrils of hamsters in groups 1 and 2 under brief but deep Isofluran^®^ gas anesthesia to prevent sneezing and coughing. Each animal received 50 µL of 3 × 10^6^ infectious units of the inoculum into each nostril. None of the animals sneezed or coughed. 

Group 3 (G3) (n = 5), epi-ocular exposed. Animals were epi-ocularly infected with 3 × 10^6^ SARS-CoV-2 viruses under Isofluran^®^ anesthesia (5%/95% oxygen) on day 0 and euthanized after 21 days.

Group 4 (G4) (n = 7), epi-ocularly exposed to challenge. Animals were epi-ocularly infected with 3 × 10^6^ SARS-CoV-2 viruses under Isofluran^®^ anesthesia (5%/95% oxygen) on day 0. After 21 days, these animals were exposed to a new intranasal infection with 3 × 10^6^ SARS-CoV-2 viruses at the same dose (challenge) and euthanized 7 days after challenge at day 28.

The inoculum SARS-CoV-2 virus (strain UCH Liège [9]) was instilled on the eyes of hamsters in the “challenge” groups 2 and 4 under brief but deep Isoflurane^®^ gas anesthesia to avoid straying by blinking. Each animal received 25 µL of 3 × 10^6^ infectious units of the inoculum to each eye.

All animals were maintained in an upright position by the investigator until they awoke (approximately 60 s after completion of intranasal or epi-ocular instillation).

### 2.2. Virus Source, Handling and Quantity

Each inoculum was obtained from a stock solution of the supernatant of a Vero cell culture (clone E6) previously exposed to SARS-CoV-2 virus (strain UCH Liège), testing 106 TCID50/mL on the same cells. The virus originated from a patient treated at the nearby University Hospital in March 2020.

### 2.3. Ante-Mortem Assessment

#### 2.3.1. Clinical Scoring

A standardized clinical examination of the animals was performed daily in a semiquantitative manner, using the scale provided by the University Animal Welfare Committee, Table 1 was used: a total score in the range of 7 to 12 for more than 24 h was provided as a reason for the decision to euthanize. None of the animals reached these values, inasmuch as no animal was so ill that such a decision was necessary. All animals survived the infections.

#### 2.3.2. Weight Gain/Deficit

The body weight of the animals was measured daily before inoculation (starting 3 days before) and twice daily thereafter (until day 14 after inoculation).

### 2.4. Plethysmography

#### 2.4.1. Overview

Lung function was measured using four single-chamber whole-body plethysmographs manufactured by EMKA Technologies^®^ (model l#PLT-UNR-RT-3). Each plethysmographic chamber was equipped with the same wire-frame pneumotachograph and differential pressure transducer (EMKA Technologies^®^, model #USB_DP_T). A hole in each chamber allowed continuous extraction of stale air at a constant flow rate (500 mL/min) and its continuous replacement with fresh air from the room using a dedicated pump (model #VENT_4_PLT). The measurement campaign for each hamster was 20 min. Before inoculation, an initial series of three manipulations was performed to familiarize the animals with the experimenter’s hands, their positioning in the device, and the plethysmographic chamber itself. Subsequent procedures were then used to record the plethysmographic flow signal generated in the plethysmographic chambers by quiet, awake breathing. The flow was systematically calibrated before and after the experiment. If a deviation of >5% was detected, the collected data were discarded and the measurement campaign was restarted.

#### 2.4.2. Analysis of Individual Breathing Patterns 

Raw flow curves were acquired by sampling the signals at 2 kHz. The regularity of the breathing pattern was first assessed manually by observing the constancy of the peak flows. Based on this criterion, with the exception of some coughing and preening, the majority of hamsters breathed regularly between the fifth and twentieth minutes in the chamber. Unlike laboratory mice, hamsters are animals that can transition from a waking state to a deep sleep state and vice versa within seconds. Because the breathing pattern differs greatly between these two states, it is critical to select episodes of regular breathing during wakefulness. On the other hand, the exploratory behavior of these animals is much more intense than in mouse or rat and is accompanied by specific breathing patterns consisting of intense sniffing. Therefore, the selection of an episode with regular breathing is not sufficient. For this reason, the selection of episodes to be analyzed must be systematically performed manually by experienced personnel. A specially designed software program (IIOX_1PULMO_4a, from EMKA Technologies^®^ (Paris, France) was then used to process the obtained flow curves. A number of parameters were measured directly based on the thoracoabdominal flow curve: inspiratory time (TI), expiratory time (TE), peak inspiratory flow (PIF), peak expiratory flow (PEF), and tidal volume (TV). The latter parameter was systematically measured twice per cycle, once during the inspiratory part (ITV) and once during the expiratory part of the respiratory cycle (ETV). Two other parameters were calculated: the time required to exhale the first 64% of TV, called the relaxation time (RT), and the bronchoconstriction index (Penh) proposed by Hamelmann and colleagues [14], Penh = [(TE/RT)−1] × (PEF/PIF). Finally, based on the parameters measured above and body weight (BW), respiratory rate [RR = 60/(TI + TE)], minute volume (MV = RR × TV), mean inspiratory flow (MIF = TV/TI), mean expiratory flow (MEF = TV/TE), and duty cycle [%TI = TI/(TI + TE)] were also calculated. Of the quantitative values obtained, the median value was systematically determined and used to calculate a mean value representative of the hamster and day in question.

#### 2.4.3. Animal Welfare

All experiments were performed in compliance with national and international animal welfare regulations. The local ethics committee of the University of Liège for animal welfare keeps the experiment under file number 20-2236.

### 2.5. Postmortem Assessment

#### 2.5.1. Autopsy

Animals were euthanized under general anesthesia (Isoflurane^®^) by neck dissection and exsanguination. During the procedure, whole blood was collected in a cryotube, coagulated at room temperature for 1–2 h, and stored at 4 °C for 24 h. It was then centrifuged, and the serum was collected in cryotubes and decomplemented at 56 °C for 45 min. The decomplemented sera were then stored at −20 °C until used for neutralizing antibody titer determination.

Necropsies were performed immediately after euthanasia by trained and experienced veterinarians. All infected animals had macroscopic lesions in both lungs that were usually multifocally distributed. The altered foci were reddish-purple in color and had a higher density than water (Docimasia-positive). No other organ exhibited morphologic changes significant enough to be detected by the naked eye. The left lung, heart muscle, liver, one kidney, spleen, and brain were carefully removed from the cadaver and placed in a 10% buffered formalin solution for 48 h. These organs were then postfixed in 70% ethanol until the day of dissection for histologic preparation. The right lung was weighed, immersed in PBS in a cryotube, immediately minced in a standardized manner (TissueLyser II, Quiagen^®^, 3 min at 30 Hz), and centrifuged. An aliquot of the supernatant was then immediately refrigerated at −80 °C for determination of infection titer (TCID50). A second aliquot was stored on RNA-later until processing for RT-qPCR.

#### 2.5.2. Histopathology and Immunohistochemistry

Tissues were subjected to the classic procedure of dehydration, paraffinization, sectioning (4 µm), and staining with hematoxylin and eosin. Lung fragments were also processed for detection of viral nucleoprotein by immunohistochemistry. Details of reagents, concentrations, and duration of each step are available upon request.

#### 2.5.3. Test for Infectious Viruses in the Lungs

The determination of infectious lung titers was performed according to the standard procedure. The main steps are summarized below:Seeding of culture plates with host cells. Seed 7.5 × 10^3^ cells per 100 µL in growth media (DMEM/FBS10%) in each well of 96-well plates. Gently swirl the plates so that the cells are evenly distributed. Grow the cells overnight. The next day, check under a light microscope to see if the cells are evenly distributed and have reached a degree of confluence of about 75%.Serial dilutions of the lung homogenate. Prepare a series of 1:10 dilutions of the lung homogenates to be titrated. The first tube is filled with 2.0 mL of infection medium (DMEM/FBS2%), and another six tubes are filled in series with 1.8 mL of infection medium. Prepare the lung homogenate with the vortex mixer and then transfer 200 μL of the suspension to the first tube. Briefly mix the first tube and prepare a 1:100 dilution by transferring 200 μL of the first to the second tube. Then transfer 200 μL of the 1:100 dilution to the next tube in the series. Repeat the procedure to make a serial 1:10 dilution of the lung suspension, e.g., from 10–1 to 10–7.Pipetting into 96-well plates. The successive 1:10 dilutions are pipetted onto the Vero cell monolayers and labeled. Provide four negative wells on each plate (which will not be in contact with the dilutions of lung homogenates). Carefully remove the growth medium from each well. Then add 100 µL of the lung dilutions per well and infect 4 wells per dilution by sweeping backwards through the dilutions. Subsequently, the cells adsorb the virus for approximately two hours at 37 °C, then 100 µL of infection medium is added to each well and the plates are placed back in the CO_2_ incubator at 37 °C or to monitor CPE for five days.Visualization and calculation of TCID50. The endpoint is reached when the CPE value appears the same on three separate readings per dilution. The titer is calculated according to the method of Reed and Muench. A titer expressed as 103 TCID/mL 50 in five days in the VeroE6 cell line can be translated as follows: 1 mL of lung homogenate diluted 1:1000 infects 50% of the cells in five days using the Vero-E6 cell line.

#### 2.5.4. Determination of Total Viral Genomic Load of the Lung

RNA was extracted from 250 µL of homogenized hamster lung using Trizol LS reagent (Invitrogen^®^, Waltham, MA, USA) according to the manufacturer’s recommendations and quantified using a Nanodrop. Subsequently, cDNA was generated by reverse transcription of 1 µg RNA using the SuperScrip IV^®^ First-Strand Synthesis System (Invitrogen^®^) with a labeled GSP according to the manufacturer’s instructions. A serial 5-point dilution of a synthetic RNA corresponding to the region of interest was reverse transcribed simultaneously. qPCRs were performed in a 20 μL volume with 1× PowerUp SYBR Green Master^®^ Mix (Applied Biosystems^®^), 500 nM of each primer, and 2 µL of cDNA. Cycling was performed with the following parameters: 2 min at 50 °C, 2 min at 95 °C, 40 cycles of 15 s at 95 °C and 30 s at 62 °C. After amplification, a dissociation curve was generated. Samples were run in triplicate.

#### 2.5.5. Quantification of the Negative Strand

RNA was extracted from 250 µL of homogenized hamster lung using Trizol LS Reagent (Invitrogen) according to the manufacturer’s recommendations and quantified using a Nanodrop. Subsequently, cDNA was generated by reverse transcription of 1 µg RNA using the SuperScrip IV First-Strand Synthesis System (Invitrogen^®^) with a labeled GSP according to the manufacturer’s instructions. A serial 5-point dilution of a synthetic RNA corresponding to the region of interest was reverse transcribed simultaneously. qPCRs were performed in a 20 μL volume with 1× PowerUp SYBR Green Master^®^ Mix (Applied Biosystems^®^), 500 nM of each primer, and 2 µL of cDNA. Cycling was performed with the following parameters: 2 min at 50 °C, 2 min at 95 °C, 40 cycles of 15 s at 95 °C and 30 s at 62 °C. After amplification, a dissociation curve was generated. Samples were run in triplicate. The Ct values obtained from the synthetic RNA were used to generate a standard curve that allows correlation between Ct value and number of molecules involved in the reaction. This curve was then used to extrapolate the number of molecules of negative SARS-CoV-2 RNA/µg lung RNA.

#### 2.5.6. Evaluation of Neutralizing Antibodies in Serum

A serum sample was collected from all hamsters to quantify neutralizing antibody titers. Virus neutralization test (VNT) was performed with SARS-CoV-2 strain BetaCov/Belgium/SartTilman/2020/1 in 96-well plates with confluent Vero E6 cells (ATCC CRL-1586). We used nine dilutions of each heat-inactivated serum (1:10 to 1:1280—corresponding to final assay dilutions 1:20 to 1:2580) so that two samples or controls could be tested per plate. For each VNT, a strong guaranteed positive control serum from the Belgian National Reference Center (Sciensano) was used. Sera were mixed vol/vol with 100 TCID/reaction50 of SARS-CoV-2 virus and incubated at 37 °C for 1 h. Then, the serum–virus mixture was transferred to the confounded cell layer in triplicate. The VNT is based on the observation of the cytopathic effect (CPE) under the light microscope at day 5 pi. Serum dilutions associated with CPE are considered negative, whereas the absence of CPE indicates complete neutralization of the SARS-CoV-2 inoculum (positive). The virus neutralization titer is reported as the highest serum dilution that neutralized CPE in 50% of wells.

## 3. Results

Animals of G2 (n = 7, exposed intranasally with challenge) and G4 (n = 7, exposed epi-ocularly with challenge) were first primed with virus by epi-ocular and intranasal infection, respectively, and then re-exposed to intranasal virus after 21 days. The experimental design was chosen to investigate whether.
epi-ocular infection with SARS-CoV-2 induces immunization with or without disease in the animal; andgeneral immunity to SARS-CoV-2 is developed after primary epi-ocular infection that protects against re-infection.

Therefore, the end point of all post-mortem analyses and plethysmographic recordings in the following chapters is the time point after the first infection (G1 and G3) or after the challenge 21 days after primary infection (G2 and G4). 

Clinically, G3 and G4 (epi-ocularly infected) did not show any clinical signs of COVID-19 disease, either localized to the eye or systemic, at any time point. The animals were industrious and not lethargic, in contrast to the animals in G1 and G2, which became ill after the first exposure to the virus, as described later. To show the difference between these groups, we give only the weight curves during the whole experiment and then compare the other curves with the groups considering only the second part of the experiment.

### 3.1. Clinical Follow-Up

#### 3.1.1. First Intranasal and Single Epi-Ocular Viral Exposure

All intranasally exposed animals assigned to groups G1 and G2 showed after initial SARS-CoV-2 exposure the COVID-19 disease. Symptoms were transient and clinically detectable from 24 h post-infection to 9–10 days post-infection with a peak of symptoms on day 5–6 post-infection. The animals became very lethargic and showed a marked decrease in locomotion, climbing, and grooming. They showed ruffled fur, sometimes hair clumped into spikes, and an almost continuous hunched back posture. The time they showed exploratory behavior (sniffing) when changing environment (from cage to plethysmograph) was reduced to almost zero. Overall, SARS-CoV-2-exposed animals from groups G1 and G2 showed progressive mean body weight loss of up to ~21% from day 1 to day 7 after infection during the first phase of primary infection and then gradually regained weight by day 14 after infection (Figure 1). 

#### 3.1.2. Epi-Ocular Virus Exposure

In contrast, animals in groups G3 and G4 initially lost minimal weight (Figure 2) but then showed steady weight gains and were lively, exploratory, and exhibited silky coats. The same was true for the second phase after epi-ocular or intranasal preconditioning 21 days later for G2 and G4. That is, after priming, they maintained their weight and activity due to immunity to the virus. Of the animals followed longitudinally, none met the Institutional Animal Care and Use Committee clinical threshold for euthanasia, of greater than 6. These data demonstrate that SARS-CoV-2 infection results in severe, nonlethal clinical signs and weight loss in animals exposed intranasally to the virus. However, epi-ocularly exposed animals did not become clinically visibly ill and apparently acquired systemic immunity as well, because they also did not become ill in the challenge experiment (analogous to intranasally exposed and recovered animals).

#### 3.1.3. Whole-Body Plethysmography

All plethysmographic data are shown in Figure 3. Penh is a composite, dimensionless value that can be used to screen experimental animals for pulmonary stress [15].

**Table 1 viruses-14-01447-t001:** Statistical analysis (*t*-test) comparing the Penh (Figure 3) of the control animals n = 24 before exposure and in the different groups after infection significant differences to the control before infection (n = 24) animals. P: level of significance and ns: not significant.

	G1-12 h Before Intransal Infection	G2-12 h Before Intransasal Re-Infection	G3-12 h Before Epiocular Infection	G4-12 h Before Intranasal Re-Infection
D3pi	*p* 0.0001	ns	*p* 0.0201	ns
D5pi	*p* 0.0001	ns	Ns	*p* 0.0102
D7pi	*p* 0.0001	ns	ns	ns
D12pi	ns	ns	ns	ns

### 3.2. Time Points

Before infection, all animals showed a similar respiratory pattern, regardless of group. After infection, two different patterns were observed. First, animals in groups G1, G2, and G4 (after the second infection) showed a pattern very similar to that before infection. There was no significant difference from the hamsters before infection. Overall, hamsters in groups G3 and G4 (after the first intranasal exposure to SARS-CoV-2) showed prolonged inhalation and exhalation on days 3, 5, and 7 after infection, which returned to normal from day 12 after infection. In these groups, mean respiratory rate, a combination of the above time points, decreased by approximately 40 cycles per minute on days 3, 5, and 7 after infection. 

### 3.3. Volumes, Mean and Peak Flow Rates

Overall, volumes and flow rates remained stable in all animals regardless of the time point studied. The general trend in hamsters of groups G3 and G4 at primo infection with SARS-CoV-2 was a dramatic increase in Penh levels on the 3rd, 5th, and 7th day post infection and a return to normal on the 12th day after infection. Flow at mid-tidal expiratory volume (abbreviated EF50) is the airflow at mid-volume of each tidal breath during expiration. EF50 remained stable throughout the study period in all infected animals. Shallow and slow flow at end-tidal expiratory volume (abbreviated EEP, standing for end-expiratory pause) is the airflow at the end volume of each tidal breath during expiration. Again, the general trend in hamsters assigned to groups G3 and G4 was an increase in EEP on days 3, 5, and 7 after infection and a return to normal by day 12 after infection. The Penh value 12 h before infection did not show significant differences between animals, so we used this value as a reference for comparison of the whole group throughout the experiment. Penh values at day 12 and 14 after infection showed no significant differences between all groups, indicating functional recovery from lung disease.

Animals in group 1 showed full disease, whereas only slight changes in Penh were observed in animals in groups 2, 3, and 4. Overall, however, there were no significant changes compared to pre-infection lung function.

### 3.4. Autopsy

Macroscopically, the lungs of the infected animals had multifocally distributed areas of red consolidations occupying between one-tenth and one-half of the surface area. The weights of the right lungs are shown in Figure 4. Here, three groups can be clearly seen. First, the lung weights of hamsters assigned to groups G4 (4.90 ± 1.16), G2 (5.41 ± 1.29), and G3 (5.20 ± 1.30) were comparable to those of healthy animals of the same age, sex, and weight (5.18 ± 0.23 mg/g body weight). In group G1 hamsters, the lungs were approximately 40% heavier.

### 3.5. Histology

Overall, the histologic changes of the lungs observed in the hamsters fell into two qualitative categories: (i) multifocal broncho-interstitial pneumonia and (ii) parenchymal multifocal interstitial pneumonia of varying intensity. In the airways, the infection was associated with marked inflammatory infiltrates and multifocal epithelial necrosis of the bronchiolar epithelium, resulting in degenerative neutrophils and cellular debris in the lumen. Within the lung parenchyma, interstitial infiltrates were systematically mixed and consisted of polymorphonuclear neutrophils, macrophages, and lymphocytes. In addition, these infiltrates were sometimes accompanied by inflammatory cells that had migrated into the alveoli (neutrophils and macrophages). Numerous hemorrhages were visible. Regeneration/repair foci (pneumocyte II hyperplasia and fibrosis) were present mainly in the peribronchial areas, suggesting an earlier inflammatory process than in the purely parenchymal areas. The (very) few immunohistochemically labeled cells were either bronchiolar epithelial cells or capillary endothelia.

Blinded attempts to divide the slides into different groups failed to reconstruct the experimental groups. In contrast, it was easy to distinguish two cohorts based on the average spatial extent of interstitial infiltrates, which were either less than 20% (cohort #1) or greater than 50% (cohort #2). Cohort #1 included all animals in groups G2, G3, and G4. Cohort #2 included the animals in group G1. Thus, at the level of histological resolution, it was possible to identify the animals from G1.

### 3.6. Viral Load in the Lungs

The density of infectious viral particles was below the detection limit in all lungs. The density of viral antigenomes (of the negative strand) in the lungs was also below the detection limit in all infected animals at the end of the respective experimental time points with clinical cure of the disease. This confirms the immunological cure.

### 3.7. Neutralizing Titers

Compared with neutralizing titers in convalescent human plasma, the titers measured here were very high (Figure 5); they were similar in hamsters assigned to the G1 once intranasally exposed and G3 once epi-ocularly exposed groups and significantly higher in the previously infected (immune) groups (G2 intranasal/challenge intranasal and G4 epi-ocular/challenge intranasal). The significance is computed in Table 2. Recovery after infection causes immunity, and re-infection (challenge) with the same virus acts as a booster. The differences between the groups G1, G3 and G2, G4 were highly significant.

## 4. Discussion

Our primary research question was whether or not an intentional epi-ocular or intranasal primary infection of hamsters with SARS-CoV-2 virus confer immunity to a second intentional intranasal inoculation with the same virus (challenge)?

In our experiments we tried to answer this question by intentional disease in groups G1 (intranasally exposed as control group), G2 (intranasally exposed with challenge), G3 (epi-ocularly exposed), and G4 (epi-ocularly exposed with challenge) and compared clinical, functional, immunological, and histological outcomes.

To obtain objective biomarkers, we observed groups G1 and G3 during the first 14 days and collected samples after euthanasia. Groups G2 and G4 were both exposed to a second infection with intranasal challenge after an epi-ocular or intranasal primary infection with 21 days of recovery, providing evidence of efficient (G2, G4) or inefficient immunization (G1, G3) via primary ocular or nasal infection. The groups G1 and G2 were sick after the primary infection intranasally. The groups G2 and G4 remained without any visible disease or weight loss. This gives proof of the infectious disease via the intranasal application and furthermore evidence of a minor disease in case of epi-ocular infection with SARS-CoV-2 virus. Evidence of induction of severe viral pneumonia in the control group (G1) is detailed above. In contrast, we observed neither weight loss nor plethysmographic changes nor lung weight gain at reinfection in the second phase of the experiment in the G2 and G4 groups after initial viral exposure (G2 intranasal and G4 epi-ocular). The histologic lesions were characteristic in nature, but minor in extent. Finally, the acquired neutralizing titers in the twice exposed groups (G2, G4) were quantitatively much higher than in the once-infected animals (G1 and G3). Thus, the humoral response was quantitatively much better after the second infection, and the challenge trial animals showed neither symptoms nor plethysmographic changes. These observations suggest that another much less pronounced viral pneumonia does occur after a second inoculation (G2 and G4) but is extremely attenuated compared with that observed after an initial infection (G1) and the first phase of intranasal infection of group G2 (not shown here). Finally, we found no differences in the challenge between the G2 and G4 groups, suggesting that the initial infection provides comparable protection regardless of the type of infection (intranasal or epi-ocular). Because the latter is without symptoms, plethysmographic changes, or lung weight gain, the epi-ocular site of application could be a very interesting option for inducing mucosal immunity to COVID-19.

This confirms results from the literature [13], which have shown that infection with SARS-CoV-2 in Syrian hamsters leads to a strong immunological response and subsequently to immunity to further infection [16]. Apparently, such a strong immunological response as in inhalational SARS-CoV-2 can also be induced by very limited, invisible conjunctival disease and mild pulmonary disease. In this case, application of the virus to the eye (epi-ocular exposure) is sufficient. This reflects findings that were inconsistent in describing the susceptibility of corneal grafts to SARS-CoV-2 virus attachment and infection, which were considered less likely [17]. In light of our experiments, a different interpretation can be presented. The MALT system around the eye efficiently processes infection in a low-inflammatory state and eliminates infectivity very early. In all animals, we did not observe ocular inflammation immediately after and during the course of epi-ocular infection. 

This opens the way to a new type of low-risk infection and possible vaccination via the conjunctiva of the eyes. This addresses several problems: if no vaccine can be found, the limited infection of one eye will yield a weak disease with full immunity, comparable to recovery.if a targeted vaccination via the eyes proves successful, the amount of vaccine administered can be reduced. In contrast to current vaccines administered by intramuscular injection, we expect to achieve not only humoral but also mucosal immunization [9].

With the work presented here, we have provided evidence that immunization via the eye is possible. A simplified vaccination procedure and a possible reduction of the amount of vaccine is thus a step closer. Similar work is currently being performed on nasal mucosa immunization with a successful approach via a nose immunization [18]. This new type of vaccination could solve many problems around the issues of sterile immunity and cost. It would likely prevent the spread of SARS-CoV-2 (and future pandemics) more efficiently.

## 5. Conclusions

The infection of the eye with SARS-CoV-2 results in subclinical disease without any weight loss or locally detectable signs of infection compared to intranasal disease. A protective immunity is acquired by ocular infection that is as effective as complete disease recovery after intranasal infection. This opens the way for a new type of vaccination against SARS-CoV-2 via the ocular conjunctiva. 

## Figures and Tables

**Figure 1 viruses-14-01447-f001:**
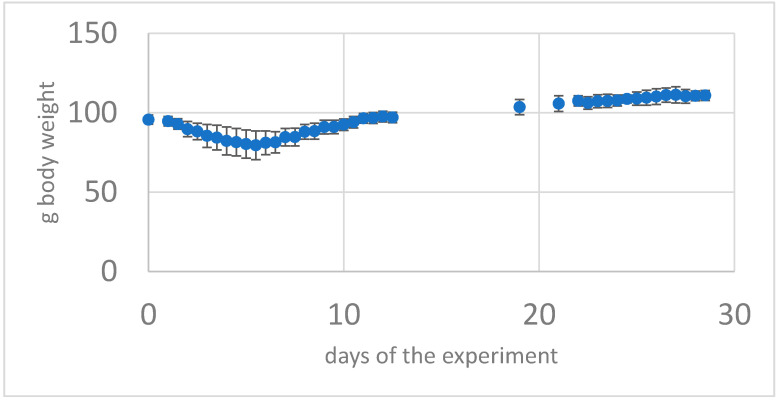
Body weight of the group G4 with a SARS-CoV-2 intranasal primary infection then 21 days later (challenge) as intranasal re-infection with SARS-CoV-2 virus (Mean of n = 7 animals + -standard deviation) From day 2 to day 8 weight loss and recovery after 12 days.

**Figure 2 viruses-14-01447-f002:**
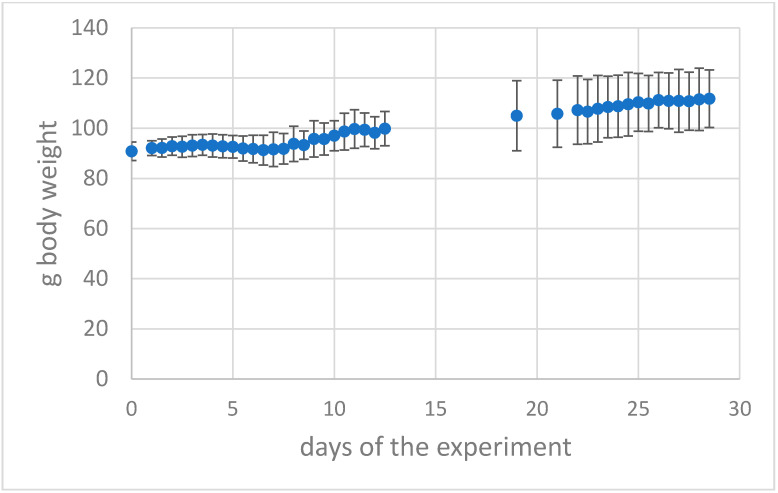
Body weight of the group G2 with a SARS-CoV-2 epi-ocular primary infection then 21 days later (challenge) with intranasal re-infection with SARS-CoV-2 virus (mean of n = 7 animals + -standard deviation). From day 4 to day 8 a slight weight loss, then recovery and increase of weight.

**Figure 3 viruses-14-01447-f003:**
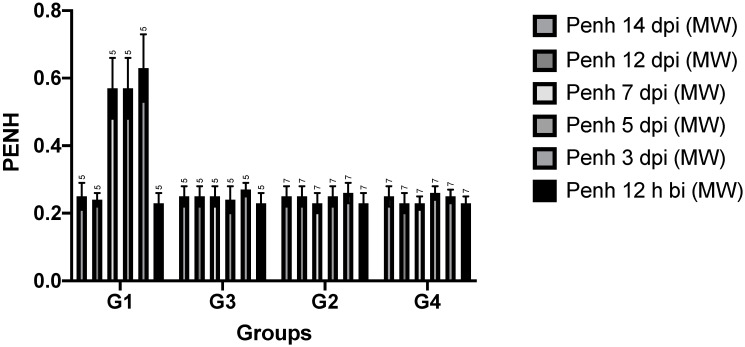
Whole body plethysmographic data as dimensionless value of Penh of all hamster groups during the experiment from 12 h before infection (bi) up to 14 days post infection (dpi) with SARS-CoV-2 virus.

**Figure 4 viruses-14-01447-f004:**
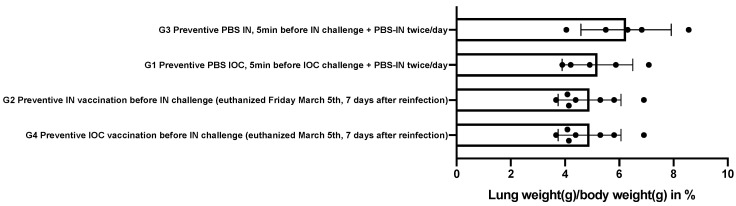
Lung weight of the right lung [g] relative to the whole body weight [g] in the different groups.

**Figure 5 viruses-14-01447-f005:**
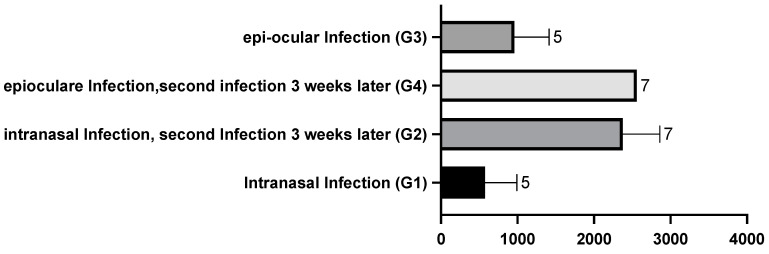
SARS-CoV-2-neutralising antibodies in the serum of the different groups of hamsters. The Y scale is in dilutions of serum resulting in a 50% mortality of the VeroE6 cells as described above. There is evidence of high antibody levels in the two previously infected and healed groups G2 and G4.

**Table 2 viruses-14-01447-t002:** This table shows that the 2 times exposure format in the healed groups lead to significantly higher neutralizing antibody titres than the disease alone. An asymptomatic re-exposure to the virus triggers a high antibody level in the blood. Interestingly, the intranasal PBS before IN virus exposure resulted in less antibody production possibly as result of severe disease. Students *t*-test p: level of significance, ns: not significant.

Group/Way of Infection IN = Intranasal IOC = Epiocular	G4	G2	G3	G1
G4 Preventive /IOC infection	X	ns	*p* 0.0013	*p* 0.0013
G2 Preventive /IN infection		X	*p* 0.0051	*p* 0.0025
G3 preventive PBS /IOC			X	ns
G1 Preventive PBS /IN				X

## Data Availability

Not applicable.

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
