# Peer review of "Eye Infection with SARS-CoV-2 as a Route to Systemic Immunization?"

_viruses, 2022, doi:10.3390/v14071447_

Round 1
Reviewer 1 Report
1. Introduction
Page 1 Lines 33-37
Paraphrase: Please change the sentence.
Suggested sentence
Previous studies suggested that systemic immunity to severe systemic disease be induced by infection of an organ not described as the primary airway targets such as the nose, throat and lung. ( Reference)
Author Response
Dear Reviewer
We improved the introduction and tried to improve the presentation of the results. I hope this will be sufficient in your eyes.
We introduced a systematic introduction and a guide into the subject of the overall SARS_ CoV2 systematic. We did not want to enoy our readers with well known facts thus we introduced a survey of the facts on pandemics.
yours sincerely N. Schrage
Reviewer 2 Report
The present study describes evidence proved that patients with SARS-CoV-2 infection in cornea or conjuntiva, could mount a strong antibody response against the virus. The study is well organized, but introduction section is really poor, and needs to be revised and deepen with most recent literature. You need to discuss also some contradictory data, such as those of Troisi et al., Microorganism, 2022, 10(2), 347.
I suggest to change the graph typology of Figure 1 and 2. I have a question:
Is the SARS-CoV-2 infection itself to induce the antibody response or other type of concomitant coronavirus infection?
Author Response
Dear Reviewer
We tried to improve the introduction section and the presentation of the results especially we tried to improve the quality of the citations on the pandemics description. The field is quiet large but as ophthalmolgist I try to stay on what I know certain thus we all try to make the work understandable.
yours sincerely N. Schrage